# A Population-Based Study of Cardiovascular Disease Mortality in Italian Cancer Patients

**DOI:** 10.3390/cancers13235903

**Published:** 2021-11-24

**Authors:** Lucia Mangone, Pamela Mancuso, Luigi Tarantini, Mario Larocca, Isabella Bisceglia, Angela Damato, Paolo Giorgi Rossi, Alessandro Navazio, Carmine Pinto

**Affiliations:** 1Epidemiology Unit, Azienda Unità Sanitaria Locale-IRCCS di Reggio Emilia, Via Amendola 2, 42122 Reggio Emilia, Italy; pamela.mancuso@ausl.re.it (P.M.); isabella.bisceglia@ausl.re.it (I.B.); paolo.giorgirossi@ausl.re.it (P.G.R.); 2Cardiology Unit, Azienda Unità Sanitaria Locale-IRCCS di Reggio Emilia, Viale Risorgimento 80, 42121 Reggio Emilia, Italy; luigi.tarantini@ausl.re.it (L.T.); alessandro.navazio@ausl.re.it (A.N.); 3Medical Oncology Unit, Comprehensive Cancer Centre, Azienda Unità Sanitaria Locale-IRCCS di Reggio Emilia, Viale Risorgimento 80, 42121 Reggio Emilia, Italy; mario.larocca@ausl.re.it (M.L.); angela.damato@ausl.re.it (A.D.); carmine.pinto@ausl.re.it (C.P.); 4Department of Biotechnologies, University of Siena, 53100 Siena, Italy

**Keywords:** cancer therapy, cardiotoxicity, cardiovascular disease, cancer mortality

## Abstract

**Simple Summary:**

Research suggests that lengthening the prognosis in cancer patients (which is certainly a positive thing) requires greater attention to be paid to the onset of other pathologies. The increase in cardiovascular mortality in cancer patients (due to both exposure to risk factors and the side effects of cytotoxic drugs) requires greater collaboration between oncologists and cardiologists, and the integration of shared follow-up paths. Particular attention to the follow-up of cancer patients can reduce this risk: in the present study, patients presented an excessive risk of cardiovascular mortality only in the first two years from cancer diagnosis.

**Abstract:**

The present research describes 25 years of cardiovascular mortality in a cohort of patients in Northern Italy. The study included patients with malignant cancer enrolled in the period of 1996–2019, and describes cardiovascular and cancer mortality in relation to sex, age, year of diagnosis, months of survivorship, tumor site, and standardized mortality ratio (SMR). Out of 67,173 patients, 38,272 deaths (57.7%) were recorded: 4466 from cardiovascular disease (CVD) (6.6%), and 28,579 (42.6%) from cancer. The proportion of CVD death increased from 4.5% in the first two years after diagnosis, to 7.3% after more than 10 years, while the proportion of deaths from cancer decreased from 70.5% to 9.4%. The CVD SMR comparing cancer patients with the general population was 0.87 (95% CI: 0.82–0.92) in 1996–1999, rising to 0.95 (95% CI: 0.84–1.08) in 2015–2019, without differences in terms of sex or age. The risk of dying from CVD was higher compared with the general population (SMR 1.31; 95% CI: 1.24–1.39) only in the first two years after diagnosis. The trend over time underscored that CVD deaths increased in patients with breast, bladder, prostate, and colorectal cancers, and, in the more recent period, for kidney cancer and melanoma patients. Our data confirmed that cardiovascular mortality is an important issue in the modern management of cancer patients, suggesting the need for an extensive interdisciplinary approach.

## 1. Introduction

Cancer and cardiovascular diseases are leading global causes of death [1]. In Italy, 170,730 deaths from cancer (men, 94,683; women, 76,047) and 220,456 deaths from cardiovascular diseases (CVD) (men, 96,017; women, 124,439) were recorded in 2018 [2].

With the recent improvements in screening, diagnosis, and treatment of many cancers, the Italian population of cancer survivors is steadily increasing [3]. In 2020, there were 3,600,000 people living with a previous diagnosis of cancer in Italy; that number is expected to increase in the coming years [4]. Several studies showed an increased risk of CVD in cancer survivors due to either their shared lifestyle risk factors or the toxicity of cancer treatments [5,6,7].

Therefore, the cardiac health of cancer survivors is increasingly important [8,9,10], especially in consideration of the progressive aging of the population [11]. Long-term cancer survivorship care is an advancing field of research [12], and the delineation of the responsibilities of primary-care physicians (PCPs) and specialists (oncologists, cardiologists, etc.) is evolving. PCPs are largely tasked with primary prevention, and cardiologists oversee the management of CVD [13].

The aims of the study, using validated data from an Italian cancer registry, were to describe mortality from CVD in patients diagnosed with incident cancer from 1996 to 2019 and to compare their risk of dying from CVD with that of the general population.

## 2. Materials and Methods

### 2.1. Case Selection

From 1996 to 2019, 94,003 cases of infiltrating malignant tumors were diagnosed in the Province of Reggio Emilia, Italy. Excluded from the initial series were 18,108 cases of nonmelanoma skin cancers, 1012 cases of chronic myeloproliferative diseases (MMPCs), 1320 nonmalignant brain tumors, and 6400 neoplasms. The series of eligible cases, therefore, comprised 67,163 patients.

In the same period, the number of total deaths from all causes was recorded: excluding death-certificate-only cases (DCOs), 38,272 deaths from all causes were included. The causes of death were classified as CVD (heart disease, hypertension, cerebrovascular disease, atherosclerosis, or aortic aneurysm/dissection; ICD-10 I00-I99) [14], malignancy (all infiltrating malignant tumors), and other causes.

Analysis had two objectives: to describe cardiovascular and cancer mortality in relation to sex, age at diagnosis, year of cancer incidence, months from diagnosis, calendar period, and tumor site, and to compare cancer patients’ risk of dying from CVD compared with that of the general resident population, adjusting for sex and age. This project did not involve any contact with the subjects and is within the statutory aims of a cancer registry. The Reggio Emilia Cancer Registry (RE-CR) was approved by the Provincial Ethics Committee of Reggio Emilia, ref. no. 2014/0019740, on 4 August 2014.

### 2.2. Data Sources

The main information sources of the RE-CR are anatomic pathology reports, hospital discharge records, and mortality data supplemented with laboratory tests, diagnostic reports, and information from general practitioners. The RE-CR covers a population of 531,891 inhabitants. It is considered to be a high-quality cancer registry because its data are up to date (incidence data refer to up to the end of 2019), and it has a high percentage of microscopic confirmations (e.g., 98.8% for breast cancer, and 93.4% for colon cancer). The percentage of DCOs is below 0.1%.

### 2.3. Statistical Analyses

Descriptive analyses of patient characteristics with cancer were performed by number of deaths for all causes, for cardiovascular disease, and for cancer. The proportions of CVD deaths and cancer deaths were calculated by age, sex, calendar period of cancer diagnosis, years since cancer diagnosis, and cancer site; we also reported the trend of the proportions of CVD and malignancy deaths among cancer survivors by calendar period of death, overall, and by sex and for the main cancer sites. Standardized mortality ratios (SMR) and relative 95% confidence intervals of dying of CVD among cancer survivors compared with the general population were calculated for all cancers sites and for specific cancer types by dividing the number of observed deaths in cancer survivors by the expected number of deaths. The expected number of CVD deaths was calculated by multiplying the age, sex, and period-specific number of person years at risk in the cohort of cancer survivors by the age, sex, and period-specific CVD mortality rates in the general population of Reggio Emilia. Subgroup analyses were conducted separately for year of diagnosis, sex, age, and years from diagnosis. Analyses were performed using STATA 13 software (StataCorp. 2013. Stata Statistical Software: Release 13., StataCorp LLC, College Station, TX, USA).

## 3. Results

We identified 67,163 eligible cancer patients from the 94,003 registered in the RE-CR in the period of 1996–2019. Of these, deaths from all causes over the same period were 38,272 (57%): 4466 (6.6%) for CVD, and 28,579 (42.6%) for cancer (Table 1).

There were slightly more cardiovascular deaths in men, with the number increasing with age, and these were more frequent in cancer patients diagnosed in the 1990s and early 2000s. Cardiovascular mortality was higher after the first 5 years following diagnosis, especially for bladder and prostate cancers (Table 1).

As for the number of years from diagnosis, the proportion of deaths from CVD in the first two years was 5.5%, and that from cancers was 86.1%, but the trends had almost reversed after 10 years from diagnosis: 29.2 and 37.6%, respectively (Figure 1).

Consequently, in the 25 years observed, the proportion of CVD deaths among all deaths in cancer survivors diagnosed since 1996 increased from 5.2% in the period of 1996–1999 to 14.2% in the period of 2015–2019 (from 4.9 to 12.9% in men, and from 5.6 to 14.2% in women), while the proportion of deaths due to malignancies decreased from 90. 7 to 65% (from 90.6 to 67.6% in men, and from 90.7 to 65.6% in women) (Figure 2).

The trend in the proportion of cancer deaths decreased for breast, prostate, colon, bladder, melanoma, non-Hodgkin’s lymphoma, and kidney cancers; there was a slight decrease for lung and stomach cancers (Figure 3).

Regarding tumor site, there were few deaths from CVD in patients with breast, prostate, and colorectal cancers in 1996–1999; the number increased to about 50 cases (per single tumor site) recorded in 2015–2019. There was an increase in cardiovascular mortality in melanoma and kidney cancer patients starting from 2015, and a slight gradual increase for lung cancer patients over the 25 years, although this remained very limited. The trend of cardiovascular mortality increased for breast, prostate, colon, melanoma, and stomach cancers, and for non-Hodgkin’s lymphoma, and decreased for bladder, lung, and kidney cancers (Figure 4).

The sites that showed the greatest proportion of cardiovascular deaths throughout the period were the prostate and melanoma, followed by the breast, corpus uteri, and bladder. As expected, the proportion of deaths from cancer was extremely high in pancreatic, lung, and stomach cancer survivors (Figure 4). The risk of death from CVD in cancer survivors compared with that of the general population is reported in Table 2. Cardiovascular mortality presented an SMR of 0.87 (95% CI: 0.82–0.92) in 1996–1999 and 0.95 (95% CI: 0.84–1.08) in the period of 2015–2019, with no significant differences between sex or age groups. Compared with the general population, the risk of dying from CVD was higher in the first two years after cancer diagnosis (SMR 1.31; 95% CI: 1.24–1.39), decreasing afterwards, with the exception of patients with lung cancer (SMR 1.42; 95% CI: 1.25–1.61).

## 4. Discussion

The main result of the present investigation was the increase in the proportion and risk of CVD deaths among cancer survivors in the last three decades. Since the second half of the 1990s, the SMR of CVD deaths in cancer survivors increased, even if it remained below 1, i.e., lower than that in the general population. This was the consequence of a decrease in cancer deaths, and an increase in cancer survival observed in our population, confirming what was reported by others [15,16]. The increase in the proportion of CVD deaths was appreciable in almost all neoplasms amenable to effective treatment and improved cancer-related survival such as breast, colorectal, prostate, bladder, kidney, and endometrial cancers and melanoma. The burden of CVD deaths was also particularly high for cancers arising in old age, probably because of the presence of coexisting structural heart disease. This could be the case in patients with urogenital tumors and justify the significant increase in CVD deaths that we observed over time in patients with bladder and prostate cancer [17,18]. Nevertheless, although cancer and CVD often share the same risk factors, oncological treatments may worsen the cardiometabolic profile, thus favoring the onset of or accelerating CVD during follow-up [19,20,21,22,23,24]. This hypothesis is supported by the higher percentage of cardiovascular deaths, mostly in obesity-related cancers (colorectal, kidney, corpus uteri, and breast) that we registered over time [25].

In comparison with the general population, our cohort of long-term cancer survivors exhibited a higher risk of cardiovascular death in the two years after diagnosis. This supports the current recommendations for the appropriate management of cardiovascular conditions and related risk factors at baseline, during, and after cancer treatment [10]. However, after two years of follow-up, we did not find any significant difference in CVD mortality compared with that in the general population [26,27]. This discrepancy may be due to several reasons, one of which related to different applications of the coding criteria of death certificates. A study showed that, in the presence of a prevalent tumor, death certificates in Italy attribute cancer as the primary cause of death, even in cases where the role of the neoplasm in determining the cascade of events leading to death is not clear [28]. Another reason may be that, like other Mediterranean countries, Italy has a low incidence and prevalence of CVD, and the universal healthcare provided by the Italian National Health Service allows for easy access to nationwide cardiovascular prevention and care [29]. This makes it difficult to compare our data with those from other studies conducted in countries with a different epidemiological impact of CVD on the general population and different healthcare systems [29]. A third reason may be improvements in the organization of care for cancer patients that have occurred over time in the Province of Reggio Emilia, where a specific pathway, including a cardio-oncology clinic, was set up to manage patients exposed to potentially cardiotoxic cancer treatments in the last decade [30]. There is increased attention to the cardiovascular problems of cancer patients and the consequent focus of cardiologists on achieving a balance between cancer treatments, cardiovascular risk factors, and an early diagnosis of CVD [8,9,10,31,32,33].

We observed an increased risk of CVD mortality during the diagnosis and treatment phase of the disease, when cancer mortality is highest. Therefore, the observed CVD excess, despite strong competitive mortality, suggests that actual excess deaths were even higher. This phenomenon was also observed in other case series [6,34]. On the other hand, despite the decrease in the competing risk of cancer mortality after the initial years from diagnosis, the CVD mortality in our cohort of cancer survivors did not exceed that of the general population.

We did not observe any difference in terms of age or sex in CVD mortality risk, even if the proportion of CVD was higher in older patients. Thus, older cancer patients must pay more attention to CVD during their cancer pathway, and the benefits and harms of treatments should be carefully evaluated.

Analysis by cancer site showed that the epidemiological transition from a disease with a very high fatality rate to a curable disease or chronic condition occurred throughout the study period for breast, colorectal, and prostate cancer, where early diagnosis and improved treatment increased survival. This was also true for bladder and kidney cancer and melanoma. In cancer survivors for these sites, the proportion of cardiovascular deaths was almost the same as that of cancer deaths.

The main limitation of our study is that we relied on the classification of death certificates for the initial cause of death. This is strongly influenced by how filling and coding rules are adopted and interpreted, showing strong variations across countries and between regions. Our study only took into consideration the death certificates of a single province, so they should be fairly uniform. A second limitation of the study is that we could not explore the role of cancer stage; in fact, stage at diagnosis changed during the study period for many cancer sites and could be a further explanation for the reduction in the proportion of cancer deaths. Unfortunately, stage is not routinely recorded for all sites by cancer registries. Comparisons with the general population should be interpreted carefully because cancer patients are subject to strong competitive mortality due to disease progression. To maintain the descriptive nature of the analysis, we decided not to adopt a competitive risk model; we report the true burden of disease and not a “what if” in the absence of cancer mortality.

One of the strengths of our study is that it was population-based and thus included all cancer cases, with the entire resident population as comparison, over a long period of time. However, we do not have any detailed information concerning treatment, which would be crucial to understanding the excess in CVD mortality observed in the first two years after cancer diagnosis.

## 5. Conclusions

Cardiovascular disease accounts for an increasing proportion of deaths in cancer patients, particularly in patients surviving more than five years from a diagnosis of breast, colorectal, prostate, or bladder cancer, or melanoma. Our study identified the acute phase of treatment and the first two years of follow-up as the most critical period for increased risk of dying from CVD, which subsequently decreases below that of the general population, possibly due to the competitive cancer mortality and to multidisciplinary surveillance.

## Figures and Tables

**Figure 1 cancers-13-05903-f001:**
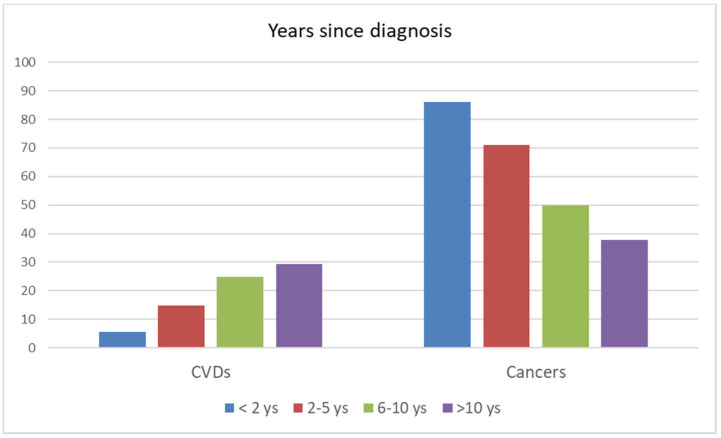
Percentage of deaths from cardiovascular disease and malignancy in cancer survivors by year of cancer diagnosis. Reggio Emilia, Italy, 1996–2019.

**Figure 2 cancers-13-05903-f002:**
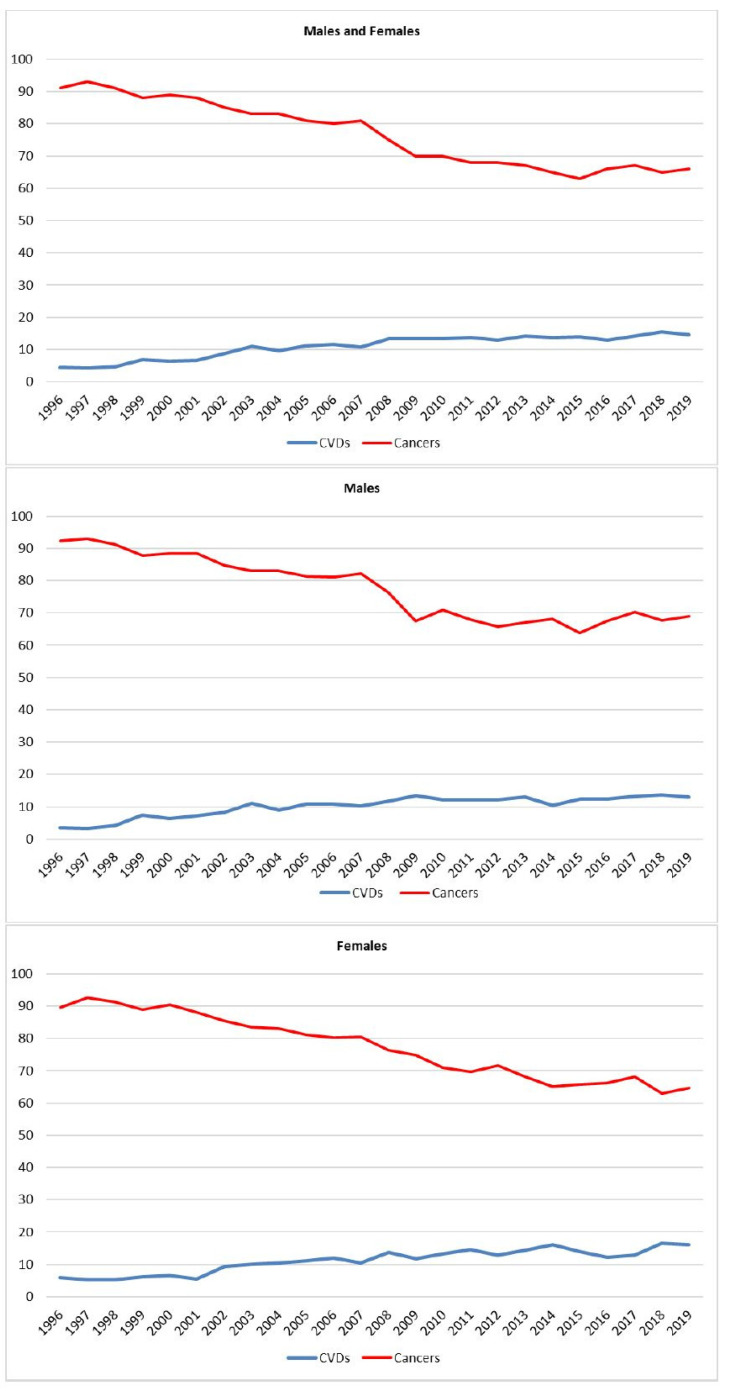
Percentage analysis of causes of death in cancer survivors diagnosed from 1996 by year of death and sex, Reggio Emilia, Italy. 1996–2019. Percentages were calculated on deaths that had occurred in the year among can-cer patients diagnosed from 1996, i.e., deaths in 1996 only include fatalities that oc-curred in patients diagnosed in 1996, while deaths that occurred in 2019 include fatali-ties in cancer survivors diagnosed from 1996 to 2019.

**Figure 3 cancers-13-05903-f003:**
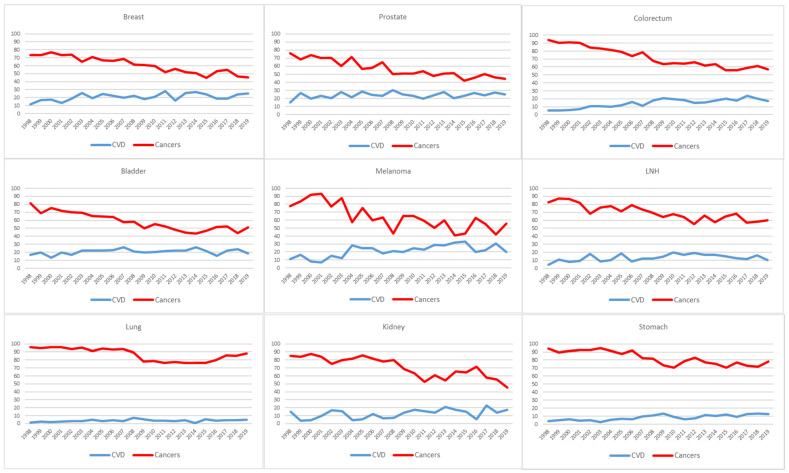
Percentage analysis of cause of death in cancer survivors by cancer site and year of death, Reggio Emilia, Italy, 1996–2019. Percentages were calculated on deaths that had occurred in the year among can-cer patients diagnosed from 1996, i.e., deaths in 1996 only include fatalities that oc-curred in patients diagnosed in 1996, while deaths that occurred in 2019 include fatali-ties in cancer survivors diagnosed from 1996 to 2019.

**Figure 4 cancers-13-05903-f004:**
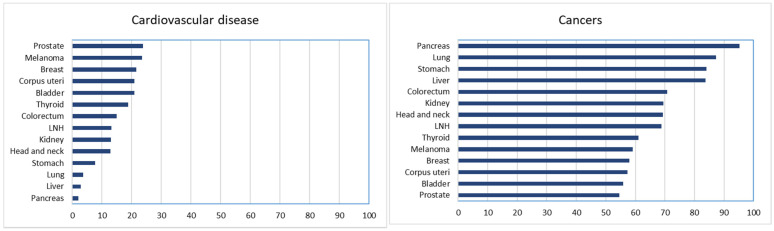
Percentage of deaths from cardiovascular disease and malignancy for 14 most important cancer sites. Reggio Emilia, Italy 1996–2019.

**Table 1 cancers-13-05903-t001:** Cancer patient incidence in Province of Reggio Emilia in the period of 1996–2019. Number and percentage of deaths, cardiovascular deaths, and cancer deaths, out of the total number of cancer patients.

	Cancer Patients	No. of Deaths		CVD Deaths		Cancer Deaths	
	No.	No.	%	No.	%	No.	%
**All**	67,163	38,272	57.0	4466	6.6	28,579	42.6
**Sex**							
Male	34,918	21,857	62.6	2445	7.0	16,345	46.8
Female	32,245	16,415	50.9	2021	6.3	12,234	37.9
**Age at diagnosis**							
0–19	425	75	17.6	0	0.0	75	17.6
20–39	2997	495	16.5	3	0.1	443	14.8
40–59	14,829	4597	31.0	122	0.8	4094	27.6
60–79	34,997	21,170	60.5	2525	7.2	15,674	44.8
80+	13,915	11,935	85.8	1816	13.1	8293	59.6
**Year of diagnosis**							
1996–1999	9965	8008	80.4	1135	11.4	5900	59.2
2000–2004	14,050	10,126	72.1	1413	10.1	7422	52.8
2005–2009	14,332	8694	60.7	1055	7.4	6346	44.3
2010–2014	14,425	6970	48.3	623	4.3	5202	36.1
2015–2019	14,391	4474	31.1	240	1.7	3709	25.8
**Year of death** *							
1996–1999	9965	3605	36.2	189	1.9	3268	32.8
2000–2004	20,410	7300	35.8	630	3.1	6230	30.5
2005–2009	27,442	8445	30.8	1012	3.7	6527	23.8
2010–2014	33,422	9051	27.1	1230	3.7	6101	18.3
2015–2019	38,762	9871	25.5	1405	3.6	6453	16.6
**Years since diagnosis**							
<2 years	27,526	22,520	81.8	1230	4.5	19,401	70.5
2–5 years	15,366	8158	53.1	1210	7.9	5789	37.7
6–10 years	11,391	4371	38.4	1085	9.5	2176	19.1
>10 years	12,880	3223	25.0	941	7.3	1213	9.4
**10 main tumor sites**							
Breast	10,295	3384	32.9	730	7.1	1963	19.1
Colorectum	7737	4668	60.3	698	9.0	3303	42.7
Lung	7306	6487	88.8	238	3.3	5668	77.6
Prostate	6452	2769	42.9	662	10.3	1511	23.4
Bladder	4438	2443	55.0	512	11.5	1365	30.8
Stomach	3298	2733	82.9	212	6.4	2298	69.7
Non-Hodgkin’s lymphoma	2679	1338	49.9	176	6.6	920	34.3
Pancreas	2343	2134	91,7	45	1.9	2034	86.8
Thyroid	2300	249	10.8	47	2.0	152	6.6
Melanoma	2038	459	22.0	108	5.2	271	13.0

* Percentages calculated on deaths that had occurred in the period among cancer patients diagnosed from 1996, i.e., deaths in 1996–1999 only include fatalities that occurred in patients diagnosed in 1996–1999, while deaths that occurred in 2000–2004 include fatalities in cancer survivors diagnosed from 1996 to 2004, and deaths in 2015–2019 include fatalities in cancer survivors diagnosed from 1996 to 2019.

**Table 2 cancers-13-05903-t002:** Standardized mortality ratio (SMR) of risk of dying from cardiovascular disease among cancer survivors compared with that of the general population. SMRs presented by year of cancer diagnosis, sex, age, and years from diagnosis. Reggio Emilia, Italy, 1996–2019.

Year of Diagnosis	No.	Expected	SMR	95% CI
1996–1999	1135	1300.64	0.87	(0.82–0.92)
2000–2004	1413	1570.77	0.89	(0.85–0.94)
2005–2009	1055	1137.27	0.92	(0.87–0.98)
2010–2014	623	677.07	0.92	(0.85–0.99)
2015–2019	240	250.67	0.95	(0.84–1.08)
**Sex**				
Male	2445	2722.58	0.89	(0.86–0.93)
Female	2021	2213.85	0.91	(0.87–0.95)
**Age**				
0–19	0	0.08	0.0	-
20–39	3	3.45	0.87	(0.28–2.69)
40–59	122	131.30	0.92	(0.77–1.10)
60–79	2525	2787.79	0.90	(0.87–0.94)
80+	1816	2013.81	0.90	(0.86–0.94)
**Years from diagnosis**				
<2 years	1230	936.68	1.31	(1.24–1.39)
2–5 years	1210	1537.98	0.78	(0.74–0.83)
6–10 years	1085	1297.39	0.84	(0.79–0.88)
>10 years	941	1164.37	0.81	(0.76–0.86)
**All cases**	4466	4936.43	0.90	(0.88–0.93)

## Data Availability

The data presented in this study are available on request from the corresponding author. The data are not publicly available due to ethical and privacy issues, requests of data should be approved by the Ethic Committee after the presentation of a study protocol.

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
