# Peer review of "A Population-Based Study of Cardiovascular Disease Mortality in Italian Cancer Patients"

_cancers, 2021, doi:10.3390/cancers13235903_

Round 1

Reviewer 1 Report

This is well conducted and described study. However, the authors needs to read carefully through the abstracts as sometimes they talk about risk of CVD, which would imply the need for SIR. The entire paper is about mortality so they need to check carefully for inconsistencies when they talk about risk of CVD death. I have no further comments as very much enjoyed reading the paper. 

Author Response

We thank the reviewer for this comment. We have corrected the mistake.

Reviewer 2 Report

The presented manuscript provides a further contribution to the already existing literature regarding mortality risk concerning cardiovascular disease in cancer patients. Thus, the study using cancer registry data can provide another component for the Real World Evidence on this topic.

The work is mainly well designed, methodologically validly implemented, and adequately discussed. However, specific points regarding methodology and discussion need to be revised.

Line 22: The abstract uses SMR without mentioning which risks or populations are being compared. Thus, an understanding of the abstract without reading the methodology in the main manuscript is not given.

Line 89: The statement "person years of risk" is quite unspecific and still needs further details for sufficient understanding (population, data source, calculation rule).

Line 103-104: The information "proportion of deaths" was not introduced in the methodology. Thus, the reader must first laboriously deduce from the subsequent text and from the figures what has been calculated here. A short section on this measure should therefore be added to the methodology (time reference: year of death, populations for numerator and denominator).

Line 146: To allow the table caption to be interpreted on its own, I recommend to add a statement to the SMR, which risks are compared with each other.

Table 2: I doubt that the risk/SMR is homogeneous with respect to cancer disease severity. Therefore, it would be interesting and relevant to show this information grouped by UICC stage.

Table 2: The expected number of deaths figure sums to the expected number for all groupings - except for years since diagnosis.  This also results in values for SMR that do not fit the other values. These numbers should be rigorously rechecked. If the numbers are correct, an explanation of how the discrepancy occurs and how it should be interpreted is required. If the information needs to be corrected, the conclusions and discussion based on it should be adapted.

Lines 155-157 and 217-218: Even if no competing risk model is calculated, the fact is very important for the interpretation of the results. Therefore, a discussion of this effect should not be omitted. If the mortality risk from cancer decreases ( i. e. longer survival over time), the proportion of other causes of death must automatically increase - even if the mortality probability for other diseases does not change. Thus, it cannot be excluded that part of the trend in the proportion of deaths is attributable to this effect alone.

Round 2

Reviewer 2 Report

Thank you for revising the manuscript closely according to the given hints. I now agree and recommend the manuscript for publication.